# The Mediating Role of Precepts and Meditation on Attachment and Depressive Symptoms in Adolescents

**DOI:** 10.3390/healthcare11131923

**Published:** 2023-07-03

**Authors:** Justin DeMaranville, Tinakon Wongpakaran, Nahathai Wongpakaran, Danny Wedding

**Affiliations:** 1Graduate School, Chiang Mai University, Chiang Mai 50200, Thailand; justinross_de@cmu.ac.th (J.D.); nahathai.wongpakaran@cmu.ac.th (N.W.); danny.wedding@gmail.com (D.W.); 2Department of Psychiatry, Faculty of Medicine, Chiang Mai University, Chiang Mai 50200, Thailand; 3Department of Clinical and Humanistic Psychology, Saybrook University, Pasadena, CA 91103, USA; 4Department of Psychology, University of Missouri-Saint Louis, St. Louis, MO 63121, USA

**Keywords:** meditation, precept, Buddhism, attachment, avoidance, anxiety, adolescent, boarding, Thai, depression

## Abstract

Research shows that Buddhist precept adherence (i.e., abstaining from killing, stealing, sexual misconduct, lying, and intoxicant use) and meditation practice influence mental health outcomes. This study investigated how Buddhist precept adherence and meditation practice influenced the relationship between insecure attachment and depressive symptoms among Thai adolescents. A total of 453 Thai boarding-school students from 10th–12th grade were recruited from five boarding schools (two purposively selected Buddhist schools and three conveniently selected secular schools). They completed these tools: Experiences in Close Relationships Questionnaire—revised-18, Outcome-Inventory-21: Depression Subscale, Precept Practice Questionnaire, and Inner-Strength-Based Inventory: Meditation. A parallel mediation model analyzed the indirect effects of attachment anxiety and attachment avoidance on depression through precept adherence and meditation practice. The participants’ demographics were 16.35 ± 0.96 years, 88% female, and 89.4% Buddhist. The mean scores for attachment anxiety were 2.7 ± 1.1; attachment avoidance, 2.78 ± 1.2; overall regular precept adherence, 20.1 ± 4.4; regular but not daily meditation, 2.94 ± 1.3; and low depressive symptoms, 3.75 ± 3.4. The standardized indirect effects for attachment anxiety (β = 0.042, 95% CI = 0.022, 0.070) and avoidance (β = 0.024, 95% CI = 0.009, 0.046) on depressive symptoms through meditation and precept adherence were significant. Meditation practice had a significantly higher indirect effect size than precept adherence.

## 1. Introduction

Attachment security originates from the internalization of support found in the infant–parent relationship [1,2,3,4,5]. A person’s attachment can be categorized as anxious or avoidant, with varying degrees of intensity [6,7]. Attachment anxiety is a hyperactivation of the attachment system and redoubling of efforts to seek support perceived as insufficient, resulting in dependent behavior. Attachment avoidance involves deactivation and suppression of the attachment system due to a perception that others are unreliable, influencing the person to be self-reliant [8]. Low scores of both anxiety and avoidance equate to secure attachment, an attachment style that enables a person to feel easily supported and capable, whereas high attachment scores equate to a disorganized attachment, which reflects an incoherent pattern of support-seeking [9]. In the absence of attachment figures, individuals can employ learned skills and coping behaviors to regulate their distress [1]. However, non-attachment-related activities are negatively influenced by the attachment system when it is activated, and this includes activities and behaviors which have positive effects. As children age into adolescence, they spend more time away from their parents. Thus, they require a broader array of adaptive skills to navigate adolescence and reduce significant mental health risks.

Insecure attachment was found to be a significant predictor for depression across studies in children [10,11], adults [12,13], and older adults [4,14]. Depression is a prevalent and disabling mental illness that affects approximately one in twenty people worldwide [15]. The COVID-19 pandemic has exacerbated depression severity in adolescents, with post-COVID-19 depression scores found to be 2.54 times higher than before the pandemic [16]. In China, the prevalence of depressive symptoms among secondary-school students during the pandemic was 28.6% [17]. Given the significant influence of attachment in the development of depression across the lifespan and the worldwide burden of depression, it is crucial to investigate existing sociocultural practices to identify and leverage available resources to address mental health challenges. 

A large minority of Thai people practice meditation (17%) and adhere to Buddhist precepts (30%) [18]. These religious practices are often used to reduce stress, a coping mechanism that can be explained by Fredrickson’s broaden-and-build theory. By calming and stabilizing individuals through non-social methods, stress-reducing behaviors can increase attachment security [19,20]. Research has shown that meditation interventions can be effective in reducing depression in secondary-school students, with an average effect size of −0.30 [21], and greatly reducing adult depressive symptoms compared to a control group after four- and eight-week interventions [22]. Dispositional mindfulness (awareness, non-judging, or acceptance of thoughts or feelings) has been found to negatively relate to attachment anxiety (r = −0.34) and avoidance (r = −0.28) [23] and to mediate between insecure attachment and maladaptive emotional regulation strategies [24]. Mindfulness meditation interventions have demonstrated that dispositional mindfulness can be increased with decreases in dissociative psychological behaviors [25]. Other research found that breathing meditation and vipassana meditation induced calmness [26] and autonomic changes [27]. Attachment insecurity can predispose individuals to maladaptive emotional regulation strategies as a means to increase perceived security [24]. Therefore, additional tools such as meditation can be employed to buffer against these predispositions and reduce the risk of depression.

The five Buddhist precepts (refrainment from killing, stealing, sexual misconduct, lying, and intoxication) are moral guidelines aimed at preventing behaviors that are considered harmful to oneself and others. Research on moral behavior indicates that morality is based in a person’s moral emotions, which are shaped by their upbringing and predispose them to prosocial behaviors [28]. In the case of insecure attachment, a person’s childhood experiences and lack of caregiver support may influence their moral compass. An insecure attachment to one’s parents can lead to a sense of insecurity about oneself and others, which may drive maladaptive behaviors such as substance abuse [29]. The five precepts are theorized to act as a buffer against such behaviors, serving as a moral ward against suicidal ideation, exploitative sexuality, dishonesty, and alcohol consumption [29,30,31,32,33,34], all of which can be risk factors for depression. A study on Thai people found that observing the precepts acted as a buffer between perceived stress and depression [35]. Combining precept adherence with meditation practice can be an adaptive technique for managing attachment-related depression. The relationship between attachment and depression has been explored with emotional regulation as a mechanism of insecure attachment’s influence on depression [36,37]. However, no research has examined the impact of lifestyle and distress regulation practices on the relationship between insecure attachment and depression. Given the widespread use and protective effects of meditation and precepts in Thailand, the authors aimed to investigate whether these practices mediate the link between insecure attachment and depressive symptoms among adolescents. Based on the positive effects of meditation and precept adherence, we hypothesized that these practices would mediate the relationship between insecure attachment and depression. 

## 2. Materials and Methods

This study analyzed data from boarding-school students in Northern Thailand between July and August of 2021. The sample consisted of males and females aged between 15 and 18 years old. The schools that participated included two urban Buddhist schools and three secular schools in urban and suburban areas, with all of the schools being in different provinces. The students practiced different meditation styles, and all were included for analysis. Special-needs students and blind or deaf students were excluded. The students provided informed consent forms, and most of the sample received parental permission. For unavailable or difficult-to-reach parents, school administrators provided consent on behalf of the parents. The Research Ethics Committee, Faculty of Medicine, Chiang Mai University, Thailand, approved this study. 

### 2.1. Instruments

Sociodemographic data were collected of each participant’s age, sex, family monthly income, religion, and school type (secular or religious). Mental health information was collected by the tools that follow.

#### 2.1.1. Experiences in Close Relationships—Revised (ECR-R-18) 

The 18-item ECR-R measures the attachment anxiety and attachment avoidance dimension with a self-rating seven-point Likert scale [38]. The Thai version has had previous clinical and non-clinical applications [39]. Originally to be used by adults about their romantic partners, the language was adapted for use by adolescents about their parents. This tool was previously found to be valid and reliable for children and adolescents across cultures [40,41]. The item scores range from 1–7, and the attachment dimension scores are calculated by reversing the avoidant attachment item scores and summing the total. Scores above the median of four are considered ‘high’ in the respective attachment dimension. The Cronbach’s alpha for this sample was 0.861 on the attachment anxiety subscale and 0.835 on the avoidance subscale.

#### 2.1.2. Inner-Strength-Based Inventory (I-SBI) 

This inventory measures the frequency of behaviors that are considered to be in accordance with the Buddhist ten perfections (e.g., loving-kindness, truthfulness, perseverance, generosity, morality, mindfulness, wisdom, patience, and endurance). Each strength is measured with one multiple-choice question that describes varying degrees of the corresponding behavior along a 5-point scale. The mindfulness perfection has a corresponding question about the frequency a person practices meditation, whereas the morality question asks about precept adherence frequency. The person reliability is 0.86 by Rasch analysis. The two-week test–retest score of the intraclass coefficient was 0.88 [42]. 

#### 2.1.3. Precept Practice Questionnaire (PPQ) 

The PPQ measures the frequency of a person’s five-precept adherence. The precepts are self-regulatory moral guidelines adopted by Buddhists to protect oneself and others [43], i.e., Killing: I avoid killing living things (including animals and insects); Stealing: I avoid taking someone’s belongings without permission; Sexual Misconduct: I avoid sexual misconduct; False Speech: I avoid telling lies; and Consuming Intoxicants: I avoid alcohol drinking and substance use. Each precept has one item that measures how often a person avoids the behavior, and a sixth item with three options questions their motivation for following the precepts. The precept frequency questions use a 5-point Likert scale (never, rarely, sometimes, often, and always). The sixth question assesses a person’s motivation for refraining from the precepts, i.e., amotivation (I have no idea why I avoid these behaviors), intrinsic (avoiding these behaviors is good for everyone), and extrinsic (I want others to see me as a good person). The precepts are calculated individually or together, ranging from 1–5 or 5–25. The motivation question can be omitted. This sample had a Cronbach’s alpha value of 0.849. 

#### 2.1.4. Outcome Inventory—Depression Subscale (OI-21) 

The OI-21 is a self-rating questionnaire that measures four domains of mental health problems, i.e., depression, anxiety, somatization, and interpersonal difficulties. The four subscales can be used individually as well as calculated together for a total score. The 21 questions ask about symptom frequency over the past week with a five-point Likert range from “never” to “always”. The depression subscale has five items measuring five depression symptoms: negative view of life, hopelessness, lack of goals, depressed, suicidality. The questionnaire and subscale’s brevity enables broad use that has proven reliable in clinical and non-clinical populations. The depression subscale can be used to screen for major depression [44]. The Cronbach alpha value was found to be 0.92 in first-year medical students [45]. The Cronbach alpha value in this study was 0.764.

### 2.2. Statistical Analysis

The sample size was calculated for linear multiple regression with a medium effect size, the significance level (alpha) set to 0.05, and the power (beta) set to 0.8. The sample size required for this secondary analysis was 114, and the total sample of 453 was included. In this sample, 3% of the data were missing. Expectation Maximization (EM) imputation was used to fill in the randomly missing data. A parallel multiple-mediator model (Figure 1) per the guidelines by Hayes [46] assessed the relationship between attachment anxiety (X1), attachment avoidance (X2), and depression (Y) through meditation (M1) and precepts (M2). Structural equation modeling (SEM) was utilized to examine the model’s fitness to the data. Attachment anxiety and attachment avoidance were tested simultaneously in a parallel mediation model with SEM. Several fit indices were employed to test the model’s adequacy: the comparative-fit index (CFI) with a threshold of 0.95 or higher; the Tucker–Lewis Index (TLI), also with a threshold of 0.95 or higher; the root mean square error of approximation (RMSEA) with a threshold of less than 0.06; and the ratio χ^2^/DF, which should be less than 3 [47]. The model’s fitness was evaluated using a maximum-likelihood estimation method for covariance matrices. A total of 5000 bootstrap resamples and the product of coefficients recommended for conducting mediation analysis were employed [48]. Standardized regression coefficients and *p*-values were reported for the coefficient direct effect, and bootstrap confidence intervals were reported for conditional indirect effect pairwise contrasts. Confidence intervals that did not cross zero indicated statistical significance. Significance levels were set at *p* < 0.05. The IBM Statistical Package for the Social Sciences (SPSS 22) and Amos, version 18 (IBM Corp., Armonk, NY, USA) were used for statistical analysis.

## 3. Results

In total, 443 participants were included in the analysis after 10 outliers were excluded from the final data. The participants’ demographic information is available in Table 1.

The mean attachment scores in this population are not interpreted as high as they are below the median score of four. The depressive symptoms are low. The precepts were adhered to often overall. The mean meditation scores indicate that participants meditated often but not every day. The measurements scores are provided in Table 2.

The correlation results indicate significant associations between attachment anxiety, attachment avoidance, meditation practice, precept adherence, and depression scores. Attachment anxiety and attachment avoidance have the highest correlation scores with depression. The meditation and precept scores are negatively related to the depression scores. Correlation scores can be found in Table 3.

Meditation and precept practices partially mediated the relationship between attachment anxiety and depression (β = 0.042, *p* < 0.001) and between attachment avoidance and depression (β = 0.024, *p* = 0.007). The attachment anxiety total effect size was β = 0.414, 95% CI = 0.347, 0.478, *p* < 0.001, whereas the attachment avoidance total effect size was β = 0.181, 95% CI = 0.103, 0.262, *p* < 0.001. The attachment anxiety direct effect on depression was β = 0.372, 95% CI = 0.301, 0.441, *p* < 0.001, whereas the attachment avoidance direct effect was β = 0.157, 95% CI = 0.075, 0.239, *p* = 0.002. The attachment anxiety indirect effect through meditation was β = −0.182, 95% CI = −0.259, −0.103, *p* < 0.001, whereas the attachment avoidance indirect effect through meditation was β = −0.120, 95% CI = −0.199, −0.045, *p* = 0.011. The attachment anxiety indirect effect through precepts was β = −0.180, 95% CI = −0.272, −0.089, *p* = 0.001, whereas the attachment avoidance indirect effect through precepts was found to be non-significant β = −0.084, 95% CI = −0.191, 0.019, *p* = 0.174. The non-significance suggests that attachment avoidance does not predict precept practice. Attachment anxiety had larger effect sizes than attachment avoidance across the analysis. Insecure attachment explained 11% (0.112) of the variance found in precept adherence and 26% (0.261) of the variance found in meditation practice frequency, while insecure attachment and both mediators (total model) accounted for 34% (0.335) of the variance in the depression results. The results can be found in Table 4. 

This model had the best fitness to the data. The fit statistics are shown as follows: CFI = 1.000, TFI = 1.008, RMSEA = 0.000 (90% CI 0.000, 0.014), SRMR = 0.008, chi-square = 0.700, df = 1, *p* = 0.403. The ratio χ^2^/DF was 0.7. 

## 4. Discussion

This study aimed to investigate the mediating role of meditation practice and precept adherence in the relationship between insecure attachment and depression, and the results support our hypothesis that both factors are associated with depression and impact the effect of attachment on depression.

The relationships between insecure attachment and the mediators were negative, implying that higher insecure attachment is less likely to predict meditation practice and precept adherence. This is a novel finding, as little previous research has accounted for the influence of attachment on behavioral frequency, particularly Buddhist religious practices. Previous research has demonstrated that low levels of mindfulness and rumination, thought suppression, and poor attentional control were found to be characteristics of insecure attachment [49], factors which may reduce a person’s ability to orientate toward meditation practice and precept adherence as behaviors to increase security.

Attachment anxiety was negatively correlated with higher meditation frequency. Attachment anxiety, characterized by a desire for proximity and attention-seeking behaviors, may undermine having a meditation practice. Meditation is often a private activity, though COVID-19 lockdowns effectively stopped group meditation at the Buddhist schools in this sample. It is the case that lockdowns provided more opportunity for meditation due to isolation, yet the data reflect that the insecurely attached participants still meditated less than the securely attached participants. Attachment anxiety may prevent one from starting and continuing a meditation period due to the cognitive-emotional states that attachment anxiety arousal generates. Attachment anxiety causes an overdependence on others and a negative view of oneself as incapable [8]. Brooding on negative affect and the dampening of positive affect are two mediators previous research found that explained the relationship between attachment anxiety and depression [50], highlighting the psychological barriers in place that may limit openness to meditation.

The analysis revealed a significant negative relationship between attachment anxiety and precept adherence. The negative relationship emphasizes the difficulties people high in attachment anxiety have in regulating their behavior. High attachment anxiety is characterized as a dependent strategy that relies on others due to a lack of adaptive coping mechanisms. Research has shown that attachment anxiety is associated with maladaptive sexual motivations, such as ‘sex in order to cope’ or for ‘self-affirmation’ [31], or greater ‘sexual compliance’ [32]. These example behaviors risk contradicting the precepts on refraining from sexual misconduct or harm. People with higher attachment anxiety have been found to report having lied to a greater number of people [34] compared with those higher in attachment avoidance. The use of maladaptive behaviors by people with high attachment anxiety may help them to increase their proximity to others, yet the authors theorize that high attachment anxiety impedes an individual’s ability to adhere to precepts. Within this sample of boarding-school students, the precepts refraining from intoxicants and sexual misconduct had the highest scores, highlighting these precepts as mostly adhered to. This can also be explained by the high level of monitoring and structure in boarding-school life. The negative association between attachment anxiety and precept adherence suggests that individuals high in attachment anxiety may struggle the most to adhere to the precepts due to their tendency to utilize maladaptive behaviors driven by their attachment-related motivations.

Attachment avoidance was found to be negatively related to meditation practice, indicating those high in attachment avoidance are less likely to meditate compared with securely attached people. Meditation, particularly in group settings, may be avoided to limit interaction with others and to prevent discomfort in response to meditation-induced changes. A study found that attachment avoidance was more negatively related than attachment anxiety to the ‘observing/noticing’ and ‘describing/labeling with words’ mindfulness traits [51]. It is possible that meditation-induced mindfulness enables recognition of previously suppressed information that conflicts with a person’s higher self-appraisal. Thought suppression was theorized to be so cognitively demanding that it caused lower mindfulness amongst avoidant-attached people [49,52]. A person with high avoidant attachment compensates for unfulfilled attachment needs by viewing oneself as self-reliant and by rejecting negative aspects about the self, such as personal weaknesses or failures that have been experienced [8]. Even short periods of daily meditation (13 min) significantly increased attention and decreased negative moods amongst meditation-naïve populations [53]. It is possible that meditation is avoided to reduce upsetting the core strategy used to compensate for an unavailable attachment figure.

Interestingly, the relationship between attachment avoidance and precept adherence was found to be non-significant (β = −0.084, *p* = 0.174), suggesting that attachment avoidance does not directly influence precept adherence. It may be that the inclusion of attachment anxiety in the model accounted for a larger portion of the variance in precept adherence.

Securely attached individuals should experience fewer obstacles toward practicing meditation and adhering to precepts. Previous research has also found secure attachment to be negatively associated with psychological difficulties and internalization problems amongst Thai adolescents [54]. Secure attachment is predictive of a higher morality that can decrease the likelihood of delinquent behaviors [55,56,57] and, by extension, lapses in precept adherence. The use of meditation and precepts likely promotes greater wellbeing and activity in opposition to depressive symptoms, further enhancing the attachment security of those who practice.

Both attachment anxiety and attachment avoidance had significant indirect effects on depression through meditation and precept adherence. Overall, the mediation analysis accounted for 34% of the variance in depression. The magnitude of the total effect size of the attachment and depression relationship was reduced after accounting for the indirect effect size (mediators). The indirect effect size of meditation was larger than that of precept adherence. Notwithstanding the small effect size, precept adherence may be more impactful than meditation. Protecting the precepts requires decision-points in time (i.e., to not lie or not kill a mosquito), perhaps multiple times each day, that result in behaviors that protect or place at risk the individual. Meditation can require a longer duration of time to induce calmness or increase mindfulness, with these benefits potentially helping precept adherence. Together, both practices may help to reduce the risk of depression.

## 5. Limitations

Caution should be exercised when generalizing the findings of this analysis to boarding-school adolescents, particularly outside of Thailand. The highly structured nature of the boarding-school environment makes direct comparisons with day-school students unadvisable. Additionally, it is important to note that two of the boarding schools from these data were Buddhist schools that give emphasis to Buddhist religious values. However, this can be replicated in other schools/demographics to see whether such practices can produce the same results. Furthermore, data on the meditation history of participants were not collected beyond their practice frequency during the previous month. As the data were collected during COVID-19 pandemic lockdowns, the schools and participants experienced routine changes and COVID-related shutdowns that limit the generalizability of these results post-COVID. Finally, the cross-sectional study design may not fully capture the definitive relationships that can be established with longitudinal data.

## 6. Conclusions

This research contributes evidence that meditation and precept practices partially mediate the attachment anxiety and attachment avoidance relationships with depression. Both mediators were negatively related to attachment and depressive symptoms. The positive influence of these Buddhist practices may lessen the effect of insecure attachment on depression. This population meditated nearly every day and frequently adhered to the precepts. This analysis provides evidence that these Buddhist practices may benefit Thai adolescents. We encourage further analysis to determine the strength of these practices required to provide benefits as well as longitudinal research to explore causality.

## Figures and Tables

**Figure 1 healthcare-11-01923-f001:**
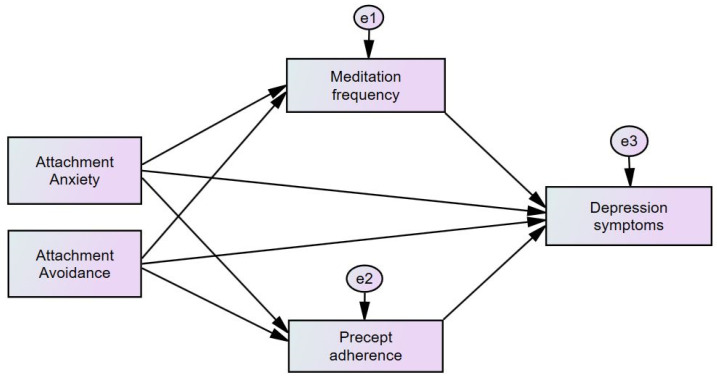
Conceptual framework of the mediation model of attachment, meditation, precept adherence, and depression. The rectangular boxes represent observed variables, single headed arrows represent paths between variables, and the circular ‘e’ represent error terms.

**Table 1 healthcare-11-01923-t001:** Sociodemographic Characteristics of the Participants.

Variables	N or Mean ± SD	%
Age	16.35 ± 0.96	-
Sex, female	390	88
School type		
Buddhist	236	53.4
Secular	207	46.6
Religion		
Buddhism	396	89.4
Non-Buddhist	47	10.6
Income (month)		
Family income—less than USD 295 *	237	53.9
Family income—USD 296 and higher	206	46.1

* 1 USD = 32 THB (exchange rate at time of study), SD = standard deviation.

**Table 2 healthcare-11-01923-t002:** Descriptive Characteristics of Measurements.

ECR-R: Attachment	N (%) or Mean ± SD	Skew	Kurtosis
Anxiety score	2.73 ± 1.1	0.473	−0.328
Avoidance score	2.78 ± 1.2	0.860	0.627
ISBI—Meditation score	2.94 ± 1.4	0.151	−1.292
ISBI—Meditation frequency			
0—I rarely meditate or have never before	73 (16.5)		
1—I try to meditate on occasions	134 (30.2)		
2—I often meditate but not every day	64 (14.4)		
3—I meditate daily at a certain time	92 (20.8)		
4—I meditate daily multiple times if able	80 (18.1)		
Precept Practice Questionnaire	20.07 ± 4.43	−1.325	1.262
Refrain from harm	3.69 ± 1		
Refrain from steal	4.1 ± 1.2		
Refrain from sexual misconduct	4.3 ± 1.2		
Refrain from lying	3.6 ± 1		
Refrain from intoxicant use	4.3 1.2		
Depression Subscale	3.75 ± 3.4	0.717	−0.387

ISBI = inner-strength-based inventory, ECR-R = experiences in close relationships—revised, SD = standard deviation.

**Table 3 healthcare-11-01923-t003:** Zero-order correlations between variables.

Variables	1	2	3	4	5	6	7	8	9	10
Age	-	0.039	−0.158 **	−0.110 *	−0.123 **	−0.065	−0.070	−0.021	−0.033	−0.027
Sex (male)		-	0.214 **	0.059	0.192 **	−0.050	−0.108 *	0.133 **	0.124 **	−0.064
Income			-	0.189 **	0.247 **	−0.127 **	−0.116 *	0.185 **	0.188 **	−0.202 **
Religion (Buddhist)				-	0.045	−0.118 *	−0.028	0.219 **	0.052	−0.119 *
School Type (Buddhist)					-	−0.154 **	−0.306 **	0.401 **	0.210 **	−0.212 **
Attachment Anxiety						-	0.385 **	−0.300 **	−0.248 **	0.509 **
Attachment Avoidance							-	−0.296 **	−0.206 **	0.371 **
Meditation								-	0.192 **	−0.335 **
Precepts									-	−0.255 **
Depression										-

* *p* < 0.05, ** *p* < 0.01.

**Table 4 healthcare-11-01923-t004:** Direct and indirect effects of insecure attachment on depressive symptoms.

Model Pathways	Coefficient (β)	SE	LL 95% CI	UL 95% CI
Total Effects (c)				
Avoidance → Depression	0.181	0.048	0.103	0.262
Anxiety → Depression	0.414	0.040	0.347	0.478
Avoidance → Meditation	−0.120	0.047	−0.199	−0.045
Avoidance →Precepts	−0.084	0.064	−0.191	0.019
Anxiety → Meditation	−0.182	0.048	−0.259	−0.103
Anxiety → Precepts	−0.180	0.055	−0.272	−0.089
Meditation → Depression	−0.143	0.043	−0.216	−0.073
Precepts → Depression	−0.085	0.047	−0.163	−0.009
Indirect Effect				
Avoidance → Depression	0.024	0.011	0.009	0.046
Anxiety → Depression	0.042	0.015	0.022	0.070
Direct Effect (c’)				
Avoidance → Depression	0.157	0.049	0.075	0.239
Anxiety → Depression	0.372	0.043	0.301	0.441

SE = standard error, LL = lower level, UL = upper level, CI = confidence interval, β = standardized coefficients.

## Data Availability

The data presented in this study are available on request from the corresponding author.

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
