# Peer review of "The Mediating Role of Precepts and Meditation on Attachment and Depressive Symptoms in Adolescents"

_healthcare, 2023, doi:10.3390/healthcare11131923_

Round 1

Reviewer 1 Report

First of all I would like to thank you for this opportunity to review this article on structural equations. 

I believe that both the introduction and the discussion should be improved and updated with more recent citations. The citation of Chen (2022) (page 2) should be corrected. Also add the initial research hypotheses at the end of the introduction. 

Regarding the structural equation analysis these present a good fit. In this case, if the variables Precept Adherence, Mediation Frequency and Depression Symptons play the role of endogenous variables (they receive effects from other variables), why has the error derived from this process not been added to the model?

Author Response

Dear Editor and reviewers

Thank you for providing us with an opportunity to improve our manuscript. Please see below our point-by-point responses to all the comments.

Reviewer 1

First of all I would like to thank you for this opportunity to review this article on structural equations.

I believe that both the introduction and the discussion should be improved and updated with more recent citations. The citation of Chen (2022) (page 2) should be corrected. Also add the initial research hypotheses at the end of the introduction.

Author Response: Thank you for your feedback. Regarding improving the intro and discussion, edits have been made. The citation for (Chen, 2022) was revised and additional citations were added. The hypothesis can be found at the end of the introduction.

Regarding the structural equation analysis these present a good fit. In this case, if the variables Precept Adherence, Mediation Frequency and Depression Symptons play the role of endogenous variables (they receive effects from other variables), why has the error derived from this process not been added to the model?

Author Response: Thank you for this keen observation. Yes, definitely, the error terms have to exist in the model. We have added the error terms to make the model more complete.

We hope that all your comments and concerns have been satisfactorily addressed. We are looking forward to hearing from you soon.

Best regard,

TW

Reviewer 2 Report

Introduction:

- The current writing gives an impression that this study is looking at Buddhism-specific phenomenon. But the survey used appears to be about general positive attitude/values and mindfulness. So it would be useful to make some revisions accordingly if relevant

- Also, if the study is looking at mental health in general (i.e., if that was the reason for using the OI), then relevant papers about other dimensions such as anxiety should be added, in addition to depression. Alternatively, please consider explaining, why was depression chosen over anxiety/other dimensions as the focus

Method

- Please consider elaborating, whether the I-SBI measures Buddhist percepts specifically, or can it be a general survey about values. Also, does the meditation item of this survey refer specifically to Buddhist meditation, or does it to (non-religion-specific) mindfulness?

- It was mentioned that, there were four domains measured using the Outcome Inventory. Did the authors explore results on the anxiety or the other dimensions too? If so, results shall be added. Or if the study was specifically about depression, it would be useful to further justify, the reason for choosing the OI instead of a more commonly used depression measure

Results

- Please consider justifying the choice of the sample size

Discussion

- Please consider elaborating, to what extend the data/results could be related to the psychological/social impact of the COVID-19 pandemic, and whether the results can be generalized post-COVID

- Because most participants were female, could this have affected the results, or could there be a gender difference?

- Please add the data availability statement

Author Response

Dear Editor and reviewers

Thank you for providing us with an opportunity to improve our manuscript. Please see below our point-by-point responses to all the comments.

Reviewer 2

Introduction:

- The current writing gives an impression that this study is looking at Buddhism-specific phenomenon. But the survey used appears to be about general positive attitude/values and mindfulness. So it would be useful to make some revisions accordingly if relevant.

Author’s Response: Thank you for your valuable feedback. Your feedback suggests that the items which measure ‘mindfulness’ and ‘morality’ appear to be general, yet both measure specific behavioral frequency relevant to Buddhist practices. The mindfulness (meditation practice frequency) and morality (precept adherence frequency) i-SBI questions are two core practices in Buddhism and represent two of the ten inner-strengths. Information was added to the i-SBI section to better inform the readers, “The mindfulness perfection has a corresponding question about the frequency a person practices meditation, whereas the morality question asks about precept adherence frequency.”

 - Also, if the study is looking at mental health in general (i.e., if that was the reason for using the OI), then relevant papers about other dimensions such as anxiety should be added, in addition to depression. Alternatively, please consider explaining, why was depression chosen over anxiety/other dimensions as the focus

Author’s Response: The relationship between depression, attachment, and the mediators is complex. This paper’s intent is to highlight how these variables relate with depression.

Method

- Please consider elaborating, whether the I-SBI measures Buddhist percepts specifically, or can it be a general survey about values. Also, does the meditation item of this survey refer specifically to Buddhist meditation, or does it to (non-religion-specific) mindfulness?

Author’s Response: Regarding the i-SBI and precepts, the question responses include ‘sila,’ which is specific to a Buddhist refrainment from behaviors for other’s and oneself, opposed to a general survey of values. The i-SBI tool and the items in question, for mindfulness/meditation and morality/precepts, capture frequency-based information about those values/attitudes and practices. The item score and total score suggests how well a person has incorporated those values/attitudes (the ten perfections) as measured behaviorally into their life. More information was provided in the instrument section that reads, “The mindfulness perfection has a corresponding question about the frequency a person practices meditation, whereas the morality question asks about precept adherence frequency.”

- It was mentioned that, there were four domains measured using the Outcome Inventory. Did the authors explore results on the anxiety or the other dimensions too? If so, results shall be added. Or if the study was specifically about depression, it would be useful to further justify, the reason for choosing the OI instead of a more commonly used depression measure

Author’s Response: Information about the OI-21 was added to the instruments section. The added content reads, “The depression subscale has five-items measuring depression symptoms: depressed mood, hopelessness, loss of interest and suicidality. The questionnaire and subscale brevity enables broad use that has proven reliable in clinical and non-clinical populations. The depression subscale can be used to screen for major depression (44).”

Results

- Please consider justifying the choice of the sample size

Author’s Response: Thank you, the power analysis was included and reads, “The sample size was calculated for linear multiple regression with a medium effect size, significance level (alpha) set to 0.05, power (beta) set to 0.8. The sample required for this secondary analysis was 114, and the total sample of 453 was included.”

Discussion

- Please consider elaborating, to what extend the data/results could be related to the psychological/social impact of the COVID-19 pandemic, and whether the results can be generalized post-COVID

Author Response: This has now been addressed in the limitations as, “As the data were collected during Covid pandemic lockdowns, the schools and participants experienced routine changes and Covid-related shutdowns that limit the generalizability of these results post-Covid.

- Because most participants were female, could this have affected the results, or could there be a gender difference?

Author Response: During the data analysis the participant sex was weighted and no difference was found between weighted and non-weighted data.

- Please add the data availability statement

Author Response: The data availability statement was added and reads, “The data presented in this study are available on request from the corresponding author.”

We hope that all your comments and concerns have been satisfactorily addressed. We are looking forward to hearing from you soon.

Best regard,

TW

Reviewer 3 Report

The Meditation article needs to be broken down some more Buddhist precept adherence and meditation practice influenced 12 the relationship between insecure a?achment and depressive symptoms"The sentence needs to be explained more in the Methods. Mindfulness as a overused construct and its basis in these practices can be explained to a broader audience (Line 73-94). Attachment can be further defined as types and explained. The reason I thought it was an exceptional article is because the authors are trying to address a difficult and abstract issues.

The limitations can be explained what the importance of a value system is. They can "this can be replicated in other schools/demographics". "Buddhist schools that give emphasis to Buddhist religious values. "

Author Response

Dear Editor and reviewers

Thank you for providing us with an opportunity to improve our manuscript. Please see below our point-by-point responses to all the comments.

Reviewer 3

1.The Meditation article needs to be broken down some more Buddhist precept adherence and meditation practice influenced 12 the relationship between insecure a?achment and depressive symptoms"The sentence needs to be explained more in the Methods. Mindfulness as a overused construct and its basis in these practices can be explained to a broader audience (Line 73-94). Attachment can be further defined as types and explained. The reason I thought it was an exceptional article is because the authors are trying to address a difficult and abstract issues.

Author Response: The meditation section of the article was developed more to include more definition in response to dispositional mindfulness and mindfulness meditation, as well as mention of other types of meditation. The section now reads, “A large minority of Thai people practice meditation (17%) and adhere to Buddhist precepts (30%) (18). These religious practices are often used to reduce stress, a coping mechanism that can be explained by Fredrickson's broaden and build theory. By calming and stabilizing individuals through non-social methods, stress-reducing behaviors can increase attachment security (19, 20). Research has shown that meditation interventions can be effective in reducing depression in secondary school students, with an average effect size of -0.30 (21), and a greater reduction in adult depressive symptoms compared to a control group after a four and eight week intervention (22). Dispositional mindfulness (awareness, non-judging, or acceptance of thoughts or feelings) has been found to negatively relate to attachment anxiety (r=-.34) and avoidance (r=-.28) (23) and to mediate between insecure attachment and maladaptive emotional regulation strategies (24). Mindfulness meditation interventions have demonstrated that dispositional mindfulness can be increased with decreases in dissociative psychological behaviors (25). Other research found breathing meditation and vipassana meditation induced calm (26) and autonomic changes (27). Attachment insecurity can predispose individuals to maladaptive emotional regulation strategies as a means to increase felt security (24). Therefore, additional tools such as meditation can be employed to buffer against these predispositions and reduce the risk of depression.”

2.The limitations can be explained what the importance of a value system is. They can "this can be replicated in other schools/demographics". "Buddhist schools that give emphasis to Buddhist religious values. "

Author Response: Thank you for this suggestion. We have revised this part.

We hope that all your comments and concerns have been satisfactorily addressed. We are looking forward to hearing from you soon.

Best regard,

TW

Round 2

Reviewer 2 Report

Thank you for the revisions.